# A SERS Study of Charge Transfer Process in Au Nanorod–MBA@Cu_2_O Assemblies: Effect of Length to Diameter Ratio of Au Nanorods

**DOI:** 10.3390/nano11040867

**Published:** 2021-03-29

**Authors:** Lin Guo, Zhu Mao, Sila Jin, Lin Zhu, Junqi Zhao, Bing Zhao, Young Mee Jung

**Affiliations:** 1State Key Laboratory of Supramolecular Structure and Materials, Jilin University, Changchun 130012, China; linguo18@mails.jlu.edu.cn (L.G.); zhulin17@mails.jlu.edu.cn (L.Z.); Zhaojq19@mails.jlu.edu.cn (J.Z.); 2School of Chemistry and Life Science, Changchun University of Technology, Changchun 130012, China; maozhu@ccut.edu.cn; 3Department of Chemistry, Institute for Molecular Science and Fusion Technology, Kangwon National University, Chuncheon 24341, Korea; jsira@kangwon.ac.kr

**Keywords:** length-to-diameter ratios, core–shell, SERS, surface plasmon resonance, au nanorods, Cu_2_O

## Abstract

Surface-enhanced Raman scattering (SERS) is a powerful tool in charge transfer (CT) process research. By analyzing the relative intensity of the characteristic bands in the bridging molecules, one can obtain detailed information about the CT between two materials. Herein, we synthesized a series of Au nanorods (NRs) with different length-to-diameter ratios (L/Ds) and used these Au NRs to prepare a series of core–shell structures with the same Cu_2_O thicknesses to form Au NR–4-mercaptobenzoic acid (MBA)@Cu_2_O core–shell structures. Surface plasmon resonance (SPR) absorption bands were adjusted by tuning the L/Ds of Au NR cores in these assemblies. SERS spectra of the core-shell structure were obtained under 633 and 785 nm laser excitations, and on the basis of the differences in the relative band strengths of these SERS spectra detected with the as-synthesized assemblies, we calculated the CT degree of the core–shell structure. We explored whether the Cu_2_O conduction band and valence band position and the SPR absorption band position together affect the CT process in the core–shell structure. In this work, we found that the specific surface area of the Au NRs could influence the CT process in Au NR–MBA@Cu_2_O core–shell structures, which has rarely been discussed before.

## 1. Introduction

Surface-enhanced Raman scattering (SERS) is a rapid, in situ, nondestructive, and ultrasensitive analytical tool. Due to the characteristics of fingerprint recognition, since its discovery in 1974 [1], SERS has been extensively studied [2,3,4,5,6]. As is well known, there are two main mechanisms for SERS enhancement: the electromagnetic mechanism (EM) and the chemical mechanism (CM) [7,8,9]. For EM mechanism, between noble metal nanoparticles (NPs), surface plasmon resonance (SPR) provides a large electromagnetic field to enable an enhancement factor (EF) up to more than 10^6^ [10,11,12]; for the CM mechanism, the enhancement comes from charge transfer (CT) process between SERS substrates and anchored molecules. Although the EF of the CM is in the range of only 10 to 10^3^, it can provide important information about the CT processes between the SERS substrate and molecules that attached on it, which is not provided by the EM mechanism. Due to the nature of the CM mechanism, it can be used for the study of many chemical processes, especially the CT process in metal–semiconductor and semiconductor–semiconductor systems [13,14,15].

As plasmonic NPs, both gold and silver each have strong localized surface plasmon resonance (LSPR) in the visible region. The SPR of noble metal nanoparticles can be further adjusted by changing the particle shape, size, and surrounding dielectric environment. Among the potential hybrid nanostructures, the metal@semiconductor core–shell heterostructure has been extensively studied because this composite material combines two completely different materials together to form a unique structure with synergistic properties and functions [16,17]. Among these materials, the Au NP@semiconductor structure is widely used in photocatalysis, solar cells, biology, sensing, and other fields [18,19,20,21,22]. Compared with Au NPs, Au nanorods (NRs) are also a commonly used SERS substrate. Au NRs have many advantages, such as two plasmon resonance bands, transverse LSPR and longitudinal LSPR [23,24], and a longitudinal LSPR absorption band that can be flexibly modulated to the required position by changing the experimental conditions. Many researchers have used Au NRs as core–shell nanostructures and semiconductors as shells to adjust the SPR of Au NRs. Au NR cores coupled with semiconductor shells are one of the most widely used structures, but few studies have been applied to SERS.

In this study, a series of Au NRs with different length-to-diameter ratios (L/Ds) were synthesized, and Au NR–4-mercaptobenzoic acid (MBA)@Cu_2_O core–shell nanostructures were synthesized. In these core–shell nanostructures, we used the same Cu_2_O shell thicknesses to obtain different SPR absorption bands by adjusting the L/Ds of Au NRs. By detecting the SERS spectra of Au NR–MBA structures and Au NR–MBA@Cu_2_O structures with 633 and 785 nm laser excitations, we obtained a series of SERS spectra that had information on the CT process in these assemblies. By analyzing the degree of CT ($\rho_{\mathrm{CT}}$), the absorption band of Au NR–MBA@Cu_2_O, and the positions of the conduction band (CB) and valence band (VB), we found that the specific surface area of the Au NRs had a certain effect on the CB and VB of the Cu_2_O shell in the Au–MBA@Cu_2_O structure. The positions of the CB and VB had a significant effect on the CT process in Au NRs and the Cu_2_O shell in the core–shell structure.

## 2. Materials and Methods

### 2.1. Chemicals

In this study, all chemicals were purchased from Sigma-Aldrich Co., Ltd. (St. Louis, MO, USA and Shanghai, China) in the highest purities available. All chemicals were applied as received without further refinement. Deionized water was used to prepare solutions we employed in this work.

### 2.2. Sample Preparation

#### 2.2.1. Preparation of Au NRs with Different L/Ds

The series of Au NRs were prepared using a seed-mediated growth method that was well-developed, with some modifications [25]. As a reducing agent, NaBH_4_ and cetyltrimethylammonium bromide (CTAB)-coated Au seeds were prepared by HAuCl_4_. CTAB solution (5 mL, 0.1 M) was primarily mixed with HAuCl_4_ solution (125 μL, 0.01 M), and ice-cold NaBH_4_ solution (0.3 mL, 0.01 M) was then added via magnetic stirring, making the solution brownish yellow. The Au seeds were prepared at 30 °C for 5 min.

Series volumes of 0.01 M AgNO_3_ solution (0.05, 0.07, 0.11, 0.15, and 0.21 mL) were added into CTAB solution (20 mL, 0.1 M) at temperature of 30 °C. Then, a HAuCl_4_ solution (1 mL, 0.01 M) was added to the AgNO_3_–CTAB solution, and after gentle stirring, ascorbic acid (AA) (0.16 mL, 0.1 M) was added into the mixed solution. The color of the seed growth solution was then gradually changed from dark yellow to colorless. Finally, 48 µL of a solution contained Au NR seeds was added into the growth solution at 30 °C, and the resulting solution was mixed and reacting for 2 h. The prepared Au NRs were centrifuged at 10,000 rpm and redispersed in deionized water, separated twice by repeating the operation.

#### 2.2.2. Preparation of Au NR–MBA with Different L/Ds

The Au NR–MBA solutions with different L/Ds were prepared in one step, adding 1 mL of 10^−5^ M MBA to 1 mL of Au NRs. The solutions were stirred for 5 min. After stirring, the mixtures were centrifuged at 10,000 rpm for 5 min, removing the supernatants.

#### 2.2.3. Preparation of Au NR–MBA@Cu_2_O with Different L/Ds

In a typical preparation of Au–MBA@Cu_2_O core–shell heterostructures with different L/Ds [26], 0.14 mL of the as-synthesized Au NR–MBA with different L/Ds were added to 5 sample vials, and a sodium dodecyl sulfate (SDS) solution (0.044 g SDS in 4.7 mL of deionized water) was introduced into the vials. Then, 0.05 mL solution of 10^−3^ M CuCl_2_, 0.125 mL solution of 1 M NaOH, and 0.075 mL solution of NH_2_OH·HCl were introduced into the reaction systems in the order listed above. NaOH solution was added during growth, the COOH group of MBA was deprotonated, Cu^2+^ and the COO^−^ group was coordinated, and a crystal nucleus was formed. With the addition of reducing agent (NH_2_OH·HCl), the Cu_2_O was grown on the surface of Au NRs, forming a core–shell structure. The solutions turned purple, and finally, as the mixture aged for 2 h, they turned to varying degrees of turquoise. To collect the products and remove the surfactant, we washed all solutions and centrifuged them 2 times with deionized water at 3500 rpm for 5 min.

### 2.3. Instruments

The ultraviolet-visible-near infrared (UV–VIS–NIR) absorption spectra were recorded by a Cary 5000 UV–VIS–NIR spectrometer (Agilent Technologies, Inc., Santa Clara, CA, USA). The surface morphology of the samples was measured by a JEOL JEM-2100 transmission electron microscopy (TEM) system (JEOL Ltd., Tokyo, Japan), operated at an acceleration voltage of 200 kV. The X-ray diffraction (XRD) patterns were obtained by a Siemens D5005 X-ray powder diffractometer (Siemens, Munich, Germany), equipped with a Cu Kα radiation source, operated at 40 kV and 30 mA. In situ Raman spectra were obtained at room temperature by a Jobin Yvon/HORIBA HR evolution Raman spectrometer (HORIBA, Ltd., Koyoto, Japan), equipped with integral BX 41 confocal microscopy. Raman spectra of MBA molecules in the Au NR–MBA@Cu_2_O system were accumulated for 30 s and 10 s in the Au NR–MBA system at room temperature. At least 5 Raman measurements were taken for each sample to verify spectral reproducibility. The spectrometer was calibrated by the Raman band of silicon at 520.7 cm^−1^.

## 3. Results and Discussion

### 3.1. Characterization of Au NRs and Au NR–MBA@Cu_2_O with Different L/Ds

TEM images of Au NRs with different L/Ds are presented in Figure 1a–e. The images clearly show the differences of L/D between these samples and its dependence of the volumes of the added AgNO_3_ solution.

From the TEM images, the average L/Ds of the Au NRs were measured and calculated as 1.96, 2.26, 2.78, 3.02, and 3.49. The size distribution of the Au NRs with different L/Ds is shown in Appendix A. TEM images of the different L/Ds of Au NR–MBA@Cu_2_O core–shell assemblies with the same Cu_2_O shells are shown in Figure 2a–e. The distribution of the Cu_2_O shell thicknesses is also shown in Figure 2. The XRD pattern for an L/D of 2.78 in the Au NR–MBA@Cu_2_O core–shell structure shown in Figure 3 shows that the core-shell structure matched the standard card of cubic Au nanocrystals (JCPDS: 04-0784, space group: Fm$\bar{3}$m, a = 0.4086 nm) and the cubic phase of Cu_2_O nanocrystals (JCPDS: 05-0667, space group: Pn$\bar{3}$m, a = 0.4269 nm). The diffraction peaks located at 38.2°, 44.5°, 64.6°, and 77.7° were assigned to the {111}, {200}, {220}, and {311} planes of the face-centered cubic Au nanocrystals. The diffraction peaks located at 2θ = 29.5°, 36.4°, 42.4°, 61.6°, 73.7°, and 77.2° were indexed to the {110}, {111}, {200}, {220}, {311}, and {222} planes of the pure cubic phase of the Cu_2_O nanocrystals, respectively. The energy-dispersive X-ray spectroscopy (EDX) spectrum and mapping results of Au NR–MBA@Cu_2_O assemblies (shown in Appendix A) suggested a successful assembly of the Au NR–MBA@Cu_2_O core–shell structure by the sandwiched MBA molecules. Additionally, no other peaks were observed, indicating a high purity of the combination (Appendix A). On the basis of the TEM images, XRD pattern, and EDX results, we can thus make the conclusion that Cu_2_O nanoshells were successfully formed, without additional phases or amorphous structures on the Au NR surfaces.

### 3.2. UV–VIS–NIR Characterization of Au NRs, Au NR–MBA, and Au NR–MBA@Cu_2_O Assemblies

UV–VIS–NIR absorption spectra of Au NRs with different L/Ds are shown in Figure 1f. These Au NRs exhibited two LSPR absorption bands of transverse and longitudinal LSPR, and also showed a similar transverse LSPR absorption band at approximately 516 nm and a longitudinal LSPR absorption band that is redshifted with increasing AgNO_3_ volume in synthesis.

The UV–VIS–NIR absorption spectra prove that Au NRs with different L/Ds have different plasmon absorption characteristics. The UV–VIS–NIR spectra, shown in Figure 4, present the absorption characteristics of Au NR–MBA assemblies, after the absorption of MBA molecules onto the Au NRs with different L/Ds. After MBA molecules were adsorbed onto the Au NRs by Au–S bonds, compared with Au NRs, the absorption of Au NR–MBA assemblies showed a slightly longer wavelength (the red lines). This comes from the result of dipole–dipole interactions between the Au NRs and MBA molecules, and the changes in the dielectric constant after the Au NRs are coated in MBA molecules. This redshift also suggests that the Au NRs combined successfully with the MBA molecules.

After the MBA molecules were attached to the Au NRs, Cu_2_O shells were grown on the Au NR–MBA assemblies via the COO^−^ bonds in the MBA molecules (shown in Figure 2f). Strong absorption was observed at wavelengths shorter than 500 nm, and this absorption was dominated by the interband transition in the Cu_2_O shell, which had exciton bands at 223, 287, and 360 nm. Au NRs with different L/Ds grew Cu_2_O shells of the same thicknesses; the similar transverse LSPR bands redshifted from 516 nm to 594 nm; and the longitudinal LSPR bands also redshifted from 615, 633, 693, 748, and 793 nm to 794, 840, 953, 1038, and 1136 nm, respectively. The plot of the SPR absorption peak position vs. the L/D of Au NRs is shown in Figure 4f.

### 3.3. SERS Spectra of MBA in Au NR–MBA and Au NR–MBA@Cu_2_O Assemblies

Figure 5c,g shows the SERS spectra of MBA molecules in different Au NR–MBA@Cu_2_O assemblies measured with 633 and 785 nm excitations. For better understanding, the SERS spectra of Au NR–MBA are also shown in Figure 5a,e. Table 1 shows the summary of Raman band assignments.

For the MBA molecules in the Au NR–MBA assemblies, two important bands attributed to a_1_-type vibrations are observed. One band at 1076 cm^−1^ was assigned to the in-plane ring breathing mode coupled with *ν*(C-S), and another band at 1584 cm^−1^ was ascribed to the totally symmetric *ν*(CC) mode. Bands at 998, 1012, and 1022 cm^−1^ were assigned to in-plane ring breathing, assignment to b_2_-type vibrations. After the coating shell of Cu_2_O was coated on the Au NR–MBA system, spectral changes were observed in these bands. These shifts are clearly described in Figure 5 and Table 1. 

It can be observed that the band at 1394 cm^−1^ (classified as the COO^−^ stretching mode) increased after the Cu_2_O shell was introduced in the system. This was because after the introduction of Cu_2_O shell, the original COO^−^ stretching mode had been promoted. At the same time, the intensity of the band at 1710 cm^−1^ (caused by the C=O stretching mode) decreased after the introduction of the Cu_2_O shell. All Raman spectra were normalized to the band at 1074 cm^−1^ in Figure 5 in order to facilitate a direct comparison. Interestingly, some changes were observed in the relative intensities of the b_2_-type vibration bands in Au NR–MBA and Au NR–MBA@Cu_2_O. Figure 5b,d,f,h show the SERS spectra in the 990–1030 cm^−1^ region to detect the relative intensity changes more clearly.

Lombardi et al. developed a CT model of SERS chemical mechanism [29,30], in which the Franck–Condon contribution can only enhance the full symmetric vibration modes of the probe molecules, while the Herzberg–Teller effect can enhance the totally symmetric and nontotally symmetric vibration modes. The CT contribution of the system is usually determined by the ratio of the non-totally symmetric vibration modes to the totally symmetric vibration modes. Lombardi and Birke defined the $\rho_{\mathrm{CT}}$ for each mode as a quantitative calculation of relative CT contribution to the intensity of SERS as
$$\rho_{\mathrm{CT}}\left(k\right)=\frac{I^{k}\left(\mathrm{CT}\right)-I^{k}\left(\mathrm{SPR}\right)}{I^{k}\left(\mathrm{CT}\right)+I^{0}\left(\mathrm{SPR}\right)}$$
where k is used as an index to determine the individual molecular lines in the Raman spectrum. Two intensities of reference lines in the spectral region without CT contributions is needed in the equation. One intensity is I^k^(SPR), that is, the intensity of the line (k) under consideration, in which only SPR contributes to the SERS intensity. The other reference is a totally symmetrical line, which is also measured by the SPR contribution, and denoted as I^0^(SPR). I^k^(CT) is the line intensity (k) measured in the spectral region, where CT resonance contributes extra to the SERS intensity.

In this study, the bands at 999 (in-plane ring breathing, b_2_) and 1182 cm^−1^ (C–H deformation modes *v*_9_, a_1_) in the Au NR–MBA@Cu_2_O system (corresponding to the bands at 998 and 1178 cm^−1^ in the Au NR–MBA system) were selected to compare the CT contributions. Different laser excitations were employed for testing the CT process.

In Figure 6a,c, the plots demonstrate the trend of the I_998_/I_1178_ ratio and $\rho_{\mathrm{CT}}$ in Au NR–MBA assemblies under 633 and 785 nm excitations. For the Au NR–MBA system, the ratio of I_998_/I_1178_ and $\rho_{\mathrm{CT}}$ tended to decrease with the increasing L/D of Au NRs under 633 nm laser excitation. Compared with Figure 4, with the increasing L/D of Au NRs, the absorption bands of the Au NR–MBA assembly deviated from 633 nm, which means that the absorption bands no longer matched the incident laser. This mismatching decreased the resonance of Au NRs and the incident laser, leading to a decrease in the CT from Au NRs to MBA molecules. At the same time, the ratio of I_998_/I_1178_ and $\rho_{\mathrm{CT}}$ simultaneously decreased. In Figure 6c, the ratio of I_998_/I_1178_ and $\rho_{\mathrm{CT}}$ increased with the increasing L/D of Au NRs. The redshift process in the absorption bands was closer to the 785 nm laser line, resulting in a stronger resonance of Au NRs and the incident laser. Intense coupling of Au NRs and the incident laser enabled prominent CT from the Au NRs to the MBA molecules, and the ratio of I_998_/I_1178_ and $\rho_{\mathrm{CT}}$ at 785 nm laser excitation showed an increasing trend. Figure 6b,d presents the trends of the ratio of I_999_/I_1182_ and $\rho_{\mathrm{CT}}$ in Au NR–MBA@Cu_2_O assemblies. In the absorption bands of Au NR–MBA@Cu_2_O assemblies shown in Figure 2f, the longitudinal LSPR absorption bands redshifted from 794 to 1136 nm, and the transverse LSPR was sustained at approximately 600 nm. As shown in Figure 6b, the ratio of I_999_/I_1182_ and $\rho_{\mathrm{CT}}$ showed a random trend at 633 nm laser excitation. In this region, the longitudinal SPR absorption bands of the assemblies far away from the laser line made a very small contribution to the SERS signal and the CT process. At the same time, the changes in transverse LSPR were not obvious and had little effect on the CT process. It is worth noting that the ratio of I_998_/I_1178_ and $\rho_{\mathrm{CT}}$ of the last sample (L/D = 3.49) increased slightly. We calculated the specific surface area of Au NRs (as shown in Appendix A) and we can see that the specific surface area of Au NRs had a tendency to increase, especially in the last sample, where the specific surface area increased sharply. This increase in the specific surface area meant an increase in the surface state, which may have been the reason for the increase in the ratio of I_998_/I_1178_ and $\rho_{\mathrm{CT}}$ for the last sample. For 785 nm laser excitation, Figure 6d shows a tendency to decrease, except for samples with an L/D of 3.49. To determine the reason for this decrease, we measured the ultraviolet photoelectron spectroscopy (UPS) of these samples.

### 3.4. UPS of Au NR–MBA@Cu_2_O Assemblies with Different L/Ds

According to the UPS results, the Fermi level of the Au NRs with L/Ds of 1.96, 2.26, 2.78, 3.02, and 3.49 corresponded to 3.94, 3.99, 4.01, 4.03, and 4.10 eV from the vacuum level, respectively (shown in Appendix A). The lowest unoccupied molecular orbital (LUMO) for MBA was located at 2.99, and the highest occupied molecular orbital (HOMO) levels were located at 7.65 eV (shown in Appendix A). The CB and VB levels of pure Cu_2_O were 5.7 and 7.9 eV, respectively (shown in Appendix A). We also used UPS to investigate Au NR–MBA@Cu_2_O assemblies with different L/Ds. Appendix A and Table 2 show the UPS results and the energy levels of CB and VB values, respectively.

The LUMO of MBA molecules (−2.99 eV) was much higher than the VB of Cu_2_O (−5.56 eV, for the highest one of the samples), which made the trend that charge transfers to the Cu_2_O from MBA molecules. In Figure 6, for 633 nm laser line (Figure 6a,b), compared with Au NR–MBA assembly, the CT degrees of samples 3, 4, and 5 (L/D 2.78, 3.02, and 3.49) in Au NR–MBA@Cu_2_O core–shell structure were significantly increased. Since LSPR absorption bands of Au NR were far away from 633 nm, the resonance effect of Au NR and laser line can be ignored, and thus the reason for the elevation of $\rho_{\mathrm{CT}}$ should be the CT from MBA molecule to the Cu_2_O. It is abnormal for samples 1 and 2 (L/D 1.96 and 2.26). In these two samples, the absorption bands of Au NR–MBA assembly were too close to the 633 nm laser line, producing a strong resonance between Au NR–MBA assemblies and the laser line, resulting in a large $\rho_{\mathrm{CT}}$. However, in Au NR–MBA@Cu_2_O assembly, LSPR absorption bands were far away from 633 nm laser line, and thus the contribution of resonance between Au NR–MBA@Cu_2_O assembly and laser line can be ignored. Therefore, samples 1 and 2 (L/D 1.96 and 2.26, respectively) were abnormal in all these samples. The same phenomenon for the same samples in laser line of 785 nm can be seen in Figure 6c,d. This was the main reason that we confirmed that the CT took place from the Au NR to MBA to Cu_2_O. In this system, according to previous studies, the CT from Au NRs to Cu_2_O shell happens in about several picoseconds, and the system becomes equilibrated. This is a dynamic equilibrium, which means the CT process occurs all the time during the excitation of laser, and flowing of electrons make the system equilibrium.

As shown in Table 2, the CB and VB values exhibited a tendency to decrease in the first four samples, but in the last sample, the CB and VB values increased suddenly, which made the VB value close to the HOMO value of MBA molecules. Theoretically, this tendency should lead to a Au NR–MBA@Cu_2_O $\rho_{\mathrm{CT}}$ value between the second sample (L/D = 2.26) and the third sample (L/D = 2.78). However, there was another factor that affected the final $\rho_{\mathrm{CT}}$ value, this being the specific surface area of Au NRs shown in Appendix A. In the last sample (L/D = 3.49), the length of Au NRs did not change significantly with increasing L/D, and in contrast, the radius of Au NRs decreased, resulting in a noticeable increase in the specific surface area of Au NRs. As the specific surface area of Au NRs increased, the volume of Au NRs decreased, which led to an increase in the CB and VB values for Cu_2_O in Au NR–MBA@Cu_2_O, making this value close to that of pure Cu_2_O. Returning to the $\rho_{\mathrm{CT}}$ value, as the specific surface area of Au NRs increased, the surface state showed a nonnegligible facilitation of the CT process, which significantly increased the $\rho_{\mathrm{CT}}$ value.

Figure 7 presents the numerical relationship between the Fermi level of Au NRs, LUMO and HOMO of the MBA molecules, and CB and VB values of the Cu_2_O shell. Arrows indicate the direction of electron transfer in the assemblies, which occurred from the Au NRs to the LUMO of the MBA molecules, and then to the CB of the Cu_2_O shell. The energy gap between the LUMO of the molecules and the CB of Cu_2_O significantly influenced the CT process. The larger the energy gap was, the more difficult the CT process. The trend of the CB value of Cu_2_O perfectly matched that of the ratio of I_999_/I_1182_ and $\rho_{\mathrm{CT}}$. The results clearly showed that the location of the CB of Cu_2_O can affect the CT process as well as the SPR position.

## 4. Conclusions

In this work, we successfully synthesized a series of Au NR–MBA@Cu_2_O complexes with different L/Ds, which each had a consistent Cu_2_O shell thickness of approximately 15 nm. We adjusted the SPR absorption band by changing the L/Ds of the core Au NRs. In these assemblies, charge can be transferred between the Au NRs and the Cu_2_O shell, with the MBA molecule acting as a bridge. To study the effect of the SPR absorption band on the CT process in metal and semiconductor core–shell structures, we measured the SERS spectra of these assemblies with 633 and 785 nm laser excitations. On the basis of an analysis of the SERS spectra of these assemblies, we can conclude that coupling of the SPR absorption band and the incident laser light had a significant effect on the CT process. The degree of CT increased with coupling of the incident laser light and the SPR absorption band. At the same time, we demonstrated that the specific surface area of Au NRs is another important factor influencing the CT process when the SPR absorption band is approximately coupled to the incident laser light. The greater the specific surface area of Au NRs, the greater the degree of CT. These results will provide a new guidance for further exploration of the CT process between metal nanoparticles and semiconductors.

## Figures and Tables

**Figure 1 nanomaterials-11-00867-f001:**
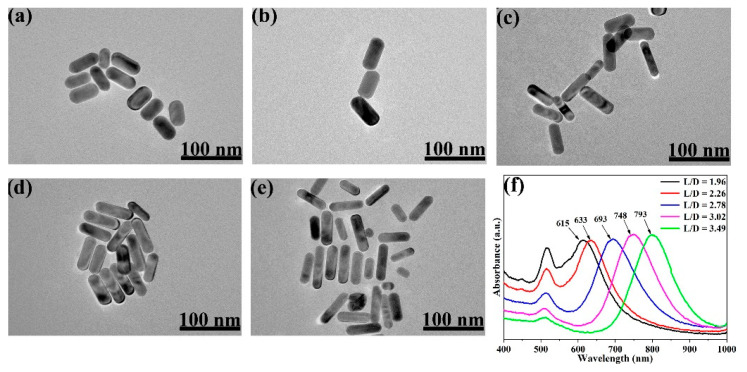
TEM characterizations of different Au nanorods (NRs) synthesized by adding series volumes of AgNO_3_: (**a**) 0.05 mL, (**b**) 0.07 mL, (**c**) 0.11 mL, (**d**) 0.15 mL, (**e**) 0.21 mL. (**f**) UV–VIS–NIR absorption spectra of Au NRs, synthesized with different volumes of 0.01 M AgNO_3_ solution (the positions of the maximum absorptions peaks are 615, 633, 693, 748, and 793 nm).

**Figure 2 nanomaterials-11-00867-f002:**
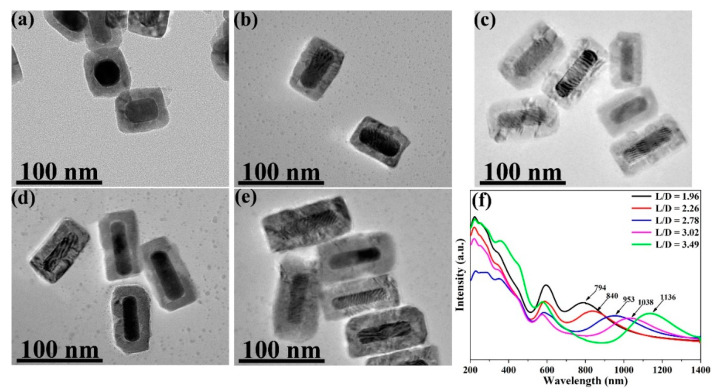
TEM images for different length-to-diameter ratios (L/Ds) ((**a**) 1.96, (**b**) 2.26, (**c**) 2.78, (**d**) 3.02, and (**e**) 3.49) of the Au NR–mercaptobenzoic acid (MBA)@Cu_2_O core–shell systems with the same Cu_2_O shell thicknesses, and (**f**) UV–VIS–NIR absorption spectra for Au NR–MBA–Cu_2_O systems with different L/D grown Cu_2_O shells of consistent thickness (the positions of maximum absorption are 794, 840, 953, 1038, and 1136 nm, respectively).

**Figure 3 nanomaterials-11-00867-f003:**
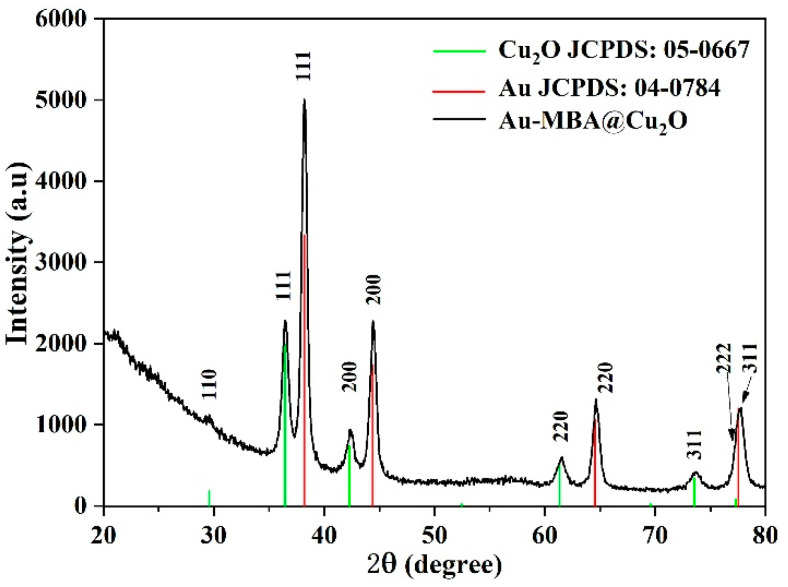
XRD pattern of the Au NR–MBA@Cu_2_O (L/D = 2.78) sample.

**Figure 4 nanomaterials-11-00867-f004:**
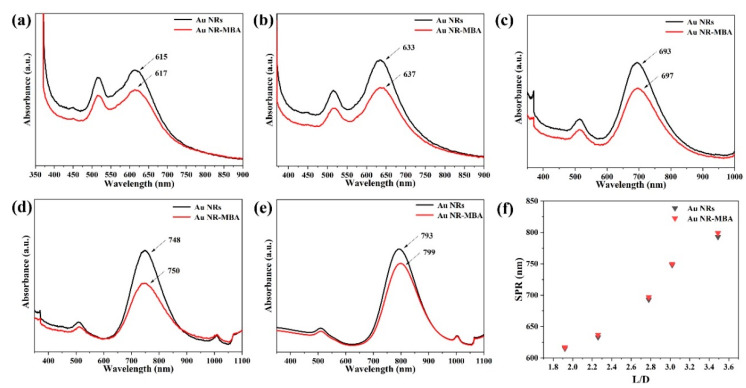
UV–VIS–NIR spectra of the Au NR–MBA assemblies with different Au longitudinal localized surface plasmon resonance (LSPR) absorption bands at (**a**) 617 nm, (**b**) 637 nm, (**c**) 697 nm, (**d**) 750 nm, and (**e**) 799 nm. (**f**) Surface plasmon resonance (SPR) absorption peak position distribution with Au NRs L/D.

**Figure 5 nanomaterials-11-00867-f005:**
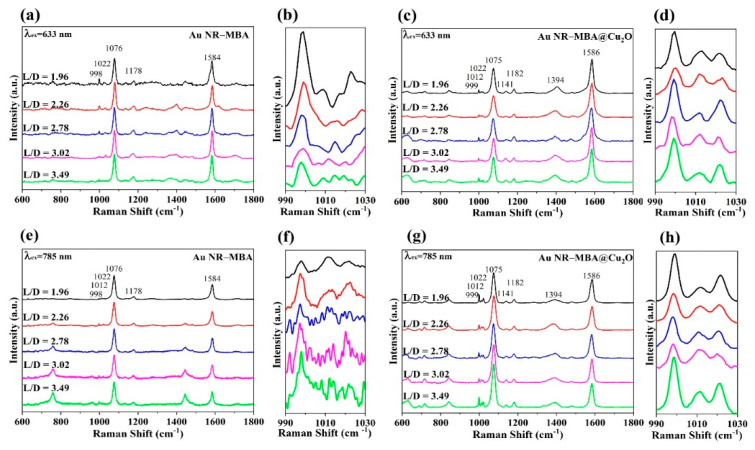
Surface-enhanced Raman scattering (SERS) spectra obtained at (**a**) 633 nm and (**e**) 785 nm laser excitations for the MBA in Au NR–MBA assemblies and SERS spectra obtained at (**c**) 633 nm and (**g**) 785 nm laser excitations for Au NR–MBA@Cu_2_O assemblies with the same shell thicknesses. Panels (**b**), (**d**), (**f**), and (**h**) are expanded views of the 990–1030 cm^−1^ regions of (**a**), (**c**), (**e**), and (**g**), respectively.

**Figure 6 nanomaterials-11-00867-f006:**
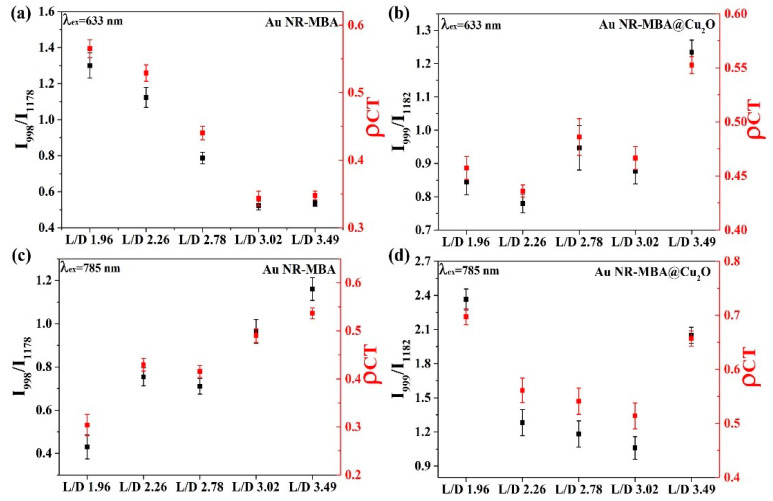
Degree of charge transfer (CT) ($\rho_{\mathrm{CT}}$) in the Au NR–MBA and Au NR–MBA@Cu_2_O samples and the SERS intensity ratio between the bands at 998 (b_2_) and 1178 cm^−1^ (a_1_) (999 (b_2_), and 1182 cm^−1^ (a_1_) of the Au NR–MBA (Au NR–MBA@Cu_2_O) assemblies at two laser excitations: (**a**,**b**) 633 nm and (**c**,**d**) 785 nm.

**Figure 7 nanomaterials-11-00867-f007:**
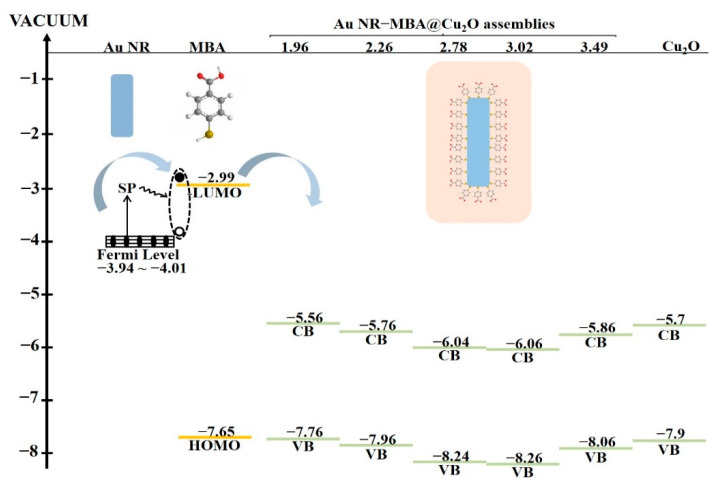
Energy level of Au NRs, MBA, Cu_2_O, and Au NR–MBA@Cu_2_O with different L/Ds.

**Table 1 nanomaterials-11-00867-t001:** Wavenumbers and bands assignments in the SERS spectrum of the MBA-modified SERS substrate [27,28].

Wavenumber (cm^−1^)		Band Assignments
Au NRs	Assemblies	
998	999	In-plane ring breathing, b_2_
1012	1012	In-plane ring breathing *+ ν*(CO), b_2_
1022	1022	In-plane ring breathing, b_2_
1076	1075	In-plane ring breathing + *ν*(C–S)
	1141	C–H deformation modes *v*_15_, b_2_
1178	1182	C–H deformation modes *v*_9_, a_1_
	1394	*ν*(COO^−^)
1584	1586	Totally symmetric *ν*(CC), a_1_
1710		C=O stretching

*ν**,* stretching. For ring vibrations, the corresponding vibrational modes of benzene and the symmetry species with C_2v_ symmetry are indicated.

**Table 2 nanomaterials-11-00867-t002:** The energy level of pure Cu_2_O and Au NR–MBA@Cu_2_O assemblies with different L/Ds.

L/D	1.96	2.26	2.78	3.02	3.49	Pure Cu_2_O
**CB (eV)**	−5.56	−5.76	−6.04	−6.06	−5.86	−5.7
**VB (eV)**	−7.76	−7.96	−8.24	−8.26	−8.06	−7.9

## Data Availability

The data presented in this study are available from the corresponding author.

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
