# Peer review of "A SERS Study of Charge Transfer Process in Au Nanorod–MBA@Cu2O Assemblies: Effect of Length to Diameter Ratio of Au Nanorods"

_nanomaterials, 2021, doi:10.3390/nano11040867_

Round 1

Reviewer 1 Report

The manuscript by Guo et al. presents the preparation and characterization of gold nanorods of different aspect ratio and their encapsulation with Cu2O. Additionally, a layer of MBA molecules was added to the Au-Cu2O interface and a detailed SERS characterization was carried out. The study is well-done and the manuscript is well-written so that I can basically recommend to publish this paper in Nanomaterials. There are some minor issues that should be addressed prior to publication:

  • Figures 6 and 7: The authors emphasize that a charge transfer takes place from the AuNR to MBA to Cu2O. How sure is it that the last step from MBA to Cu2O actually takes place? The analysis presented in figure 6 only takes the CT to MBA into account, right? How suitable is the approach proposed by Lombardi et al. for a case where the charge is not staying on a molecule, but is further transferred to a semiconductor? What are the time-scales involved? A few more explanatory sentences would help a lot.
  • Supporting Information: The authors derive the energy levels shown in Figure 7 from UP spectra shown in the SI. Please explain in more detail, how the actual values are extracted from the spectra, as this is not obvious.
  • 1, l. 37: Much higher enhancements than 10^6 can be achieved, see e.g. J. AM. CHEM. SOC. 2010, 132, 10903–10910; Nanoscale, 2016, 8, 5612–5620; J. AM. CHEM. SOC. 9 VOL. 131, NO. 40, 2009.
  • 1, l. 39: delete “mechanism”
  • 2, l. 62: adsorption should be absorption
  • 4, l. 143: successfully should be successful
  • p 5, l. 174: were adsorption should be were adsorbed
  • 6, l. 216: The authors write “…the original COO^- stretching mode has been promoted.” What does “promote” mean here? In this context, there is an additional band appearing in SERS spectra at around 1430 cm-1 (very strongly in fig. 5e), which is not addressed by the authors. Do they have an idea about its origin?
  • 7, l. 224: it should be Franck-Condon
  • 7, l. 225: I guess that it should be Herzberg-Teller instead of Herzberg-Taylor.
  • 7, l. 258: decreases should be increases, I guess.
  • 7, l. 274, 275: What is meant by “…increase in the surface state”?

Author Response

We also appreciate the Reviewers’ valuable comments and suggestions on our manuscript. According to the comments of Reviewers, we have prepared a highlighted revised version documenting all changes made. Our point-by-point responses to the editorial and Reviewers’ comments is included with this letter. 

Reviewer 2 Report

In this work, a new Au-MBA@Cu2O SERS substrate have been fabricated by wet-chemical methods where the length to diameter ratios (L/Ds) of core AuNRs varied by the variation of AgNO3 volume fraction. The morphologies, size distribution, extinction spectrum, and mechanism of charge transfer of resultant substrates was characterized in detail with TEM, XRD, UV-vis, Raman scattering and UPS. The obtained results are interesting in the relative field. However, the following points should be clarified before publication and my opinion could be considered for publication after a major revision.

  1. How do authors keep the thickness of Cu2O shell consistent with all Au-MBA@Cu2O samples with different L/Ds of AuNR?
  2. Howdo authors know the MBA successfully grafted to the surface of AuNRs?
  3. More detailneeded for the explanation of growth machanism of Cu2O
  4. As authors mentioned, “the energy-dispersive X ray-spectroscopy spectrum...” at page 4, where is the EDX spectrum? I cannot find it in the manuscript.
  5. Whatis the reason for the red shift of adsorption of peaks after attached to the surface of AuNRs?  Please explain it.  
  6. The three line format of table 1 can not meet the requirement of NANOMATERIALS, please revise it.
  7. The letter ( ρCT) in the main text is different from the letter from equation, please revise it.
  8. “In figure 6c, the ratio of ........decreases with the increasing L/D...”at page 8,  the explanation does not match the value trends in figure 6c.
  9. For the reason of CT mechanism part, more detail needed for this, and also some references could provide. Not only describe the information from the figures, but the depth analysis is also needed.

Author Response

(The authors gave the same response as above.)

Round 2

Reviewer 2 Report

Authors have revised all points as comments mentioned. This work should have some benefits for this research field. So, it should be accepted.

Author Response

We appreciate Reviewer’s comments.